# Neuronal Actin Remodeling and Its Role in Higher Nervous Activity

**DOI:** 10.3390/ijms262211215

**Published:** 2025-11-20

**Authors:** Aleksandr V. Zhuravlev

**Affiliations:** Pavlov Institute of Physiology, Russian Academy of Sciences, 199034 Saint Petersburg, Russia; beneor@mail.ru

**Keywords:** brain, actin, cofilin, memory, forgetting, mouse, Drosophila, artificial neural networks

## Abstract

The dynamic interaction of memory and forgetting processes determines the formation, stability, and specificity of the engram. While the molecular genetic processes of learning and memory have been intensively studied, the mechanisms of active forgetting have only recently attracted the attention of neuroscientists. The emergence and disappearance of memory traces in the brain are regulated by specific signaling cascades that influence the morphological and functional properties of synaptic connections. Actin remodeling is known to be the basis of neuroplasticity. Cofilin normally acts as an actin severing protein, allowing the actin cytoskeleton to locally change its structure. LIMK-dependent inactivation of cofilin stabilizes filamentous (F)-actin in dendritic spines, being crucial for engram consolidation. On the other hand, a lack of globular (G)-actin prevents actin remodeling, so inactivation of cofilin also stimulates forgetting after learning. The effects of cofilin-dependent signaling pathways on the engram depend on both the type of memory and the model object. In this review, I focus on the role of neuronal actin remodeling in learning, memory retention and forgetting processes, as well as the signal pathways that govern actin cytoskeleton dynamics. Parallels between neuroplasticity and learning in artificial neural networks (ANNs) are also discussed.

## 1. Introduction

What is the maximum memory capacity of the human brain? No one knows the correct answer. Even a rough estimate requires some theoretical model of the organization of memory traces in the brain. Donald Hebb’s learning theory states that “neurons wire together if they fire together” [1,2]. The above postulation is the basis of the so-called associative/connectionist concept of memory [3]. Engrams are thought to be embodied in a distributed pattern of neural interactions, the complex order of which determines all the properties of engram and the way it is recollected by the brain in response to an external stimulus.

According to the model of sparsely encoded associative memory, the number of independent engrams is ~N^2^logN^2^, where N is the number of neurons [4]. Estimations from various researchers give a value of ~10^15^–10^17^ or even 10^20^–10^21^ bits [5]. Such a huge memory capacity is theoretically sufficient to encode all events that happen to a person throughout life. In many cases, forgetting appears to be more of a reduction in the availability of engrams than a permanent memory loss. Thus, forgotten memory can be recovered under certain circumstances. This primarily applies to immediate memory and short-term memory (STM), but can also occur for memories of old events [6]. The main problem appears to be the ability to purposefully recall engrams. This is likely limited by the brain’s basic operating principles, as it appears unable to optimally organize engrams, making them all ready for retrieval. This would likely require some sort of «memory archive», where specific engrams are hierarchically ordered according to the place and time of their formation.

Something similar is observed in rare cases of highly superior autobiographic memory (HSAM). People with HSAM are able to effortlessly and accurately recall almost every day of their past in detail, using dates from their lifetime as cues for retrieving information. Their intelligence and the ability to memorize recent events appear to be in the normal range, but they seem to have enhanced memory consolidation and/or decreased ability to forget events that happened one or more months ago. These people also show significant changes in certain neural pathways, including those connecting the hippocampus to other areas of the brain. Greater neural activity controlling memory retrieval is observed as well. Increased memorization may also be caused by a disturbance of the brain system that estimates the relevance of information [7]. Hence, people with HSAM may differ from ordinary people both in the way they process information at the neurophysiological level and in the way they consolidate memory at the molecular level, which requires further research. Mnemonic training can organize abstract information as visuospatial routes, which also helps to retrieve it [8].

At the same time, impairment of the ability to forget information appears to be harmful in many cases. Deficits in incidental or intentional forgetting are seen in patients with attention-deficit/hyperactivity disorder, schizophrenia, obsessive–compulsive disorder, post-traumatic stress disorder, and other diseases characterized by intrusive thoughts and unwanted impulses [9]. Special molecular mechanisms of active forgetting are aimed at eliminating engrams that are no longer relevant. By competing with memory acquisition and consolidation, they provide that only specific association between two stimuli will be memorized by the brain. Therefore, the role of memory and active forgetting in cognitive performance is of an equal importance [10]. Moreover, when the brain recalls some consolidated memory, a window of opportunity opens to forget it and to form a new engram [11]. Figuratively speaking, neuronal connections are like the fused contacts of a microchip, the operation of which melts them, increasing their ability to dissociate and form a new pattern of interaction with other “hot” contacts. This dynamic competitive system of interactions allows the brain to respond flexibly to a changing environment, optimizing its work according to current tasks.

To form connections and reorganize them over different time scales, neurons have developed a complex molecular machinery, including specific cellular receptors, signal cascades, transcription factors, non-coding RNA, enzymes that modify DNA and histones to regulate gene expression, and so on [12,13]. It is hard to consider all of them within the framework of one article. Here, I focus on the neuronal actin cytoskeleton, which plays a special role in learning and memory. The forms of actin in neurons and the signal pathways that regulate its remodeling, as well as the formation, maintenance, and erasure of engrams, are considered. In the Section 8, I draw parallels between natural and artificial intelligence, attempting to answer the question of what makes the brain so efficient at processing complex information.

## 2. Organization of Actin Cytoskeleton in Neurons

Actin and its regulatory factors are the most abundant protein group in the cell. Mammals and birds have six actin genes, including *α_skeletal_*, *α_cardiac_*, *a_smooth_*, and *γ_smooth_*, which are expressed predominantly in skeletal muscle, cardiac muscle, and smooth muscle, respectively, while *β_cyto_* and *γ_cyto_* are expressed ubiquitously [14]. In *Drosophila melanogaster*, *act5C (β-actin)* and *act42A* are expressed in the cytoplasm of undifferentiated cells, *act57A* and *act87E* predominantly in intersegmental muscles, *act88F* in the thorax, and *act79B* in the thorax and leg muscles [15].

In the cell, actin exists either in monomeric form (G-actin) or in polymerized form (F-actin). F-actin is a double-stranded helical polymer with a diameter of about 7 nm, having a barbed (plus) and pointed (minus) ends. F-actin can polymerize from ATP- or ADP-bound G-actin in three stages: nucleation, elongation, and steady state. ATP-bound actin subunits are added to the faster-growing barbed end, followed by rapid hydrolysis of ATP to ATP-P_i_ and slow dissociation of P_i_, which significantly weakens ADP-actin affinity to the filament. Finally, ADP-actin is released from the slower-growing pointed end [16]. The process of movement of G-actin subunits along the filament, known as treadmilling, is essential for directional movement of neuron outgrowths. Enhanced pointed-end depolymerization of F-actin increases the stationary concentration of G-actin, which in turn promotes the barbed end growth [17].

Actin behavior is regulated by various protein factors, including actin monomer-binding proteins, severing proteins, nucleation proteins, actin filament polymerases, capping proteins, cross-linking proteins, and filament-binding proteins [18,19,20]. In the cell, nucleation of actin is the rate limiting stage of its polymerization. Therefore, it must be precisely regulated by specific factors to ensure F-actin assembly at exactly the right place and time. While formin promotes the formation of unbranched filaments, the Arp2/3 complex after activation by WASP nucleates branched filaments. The fast growing barbed ends are often oriented towards the cell membrane. Actin capping proteins, such as a conservative capping protein (CP), bind to the barbed end, blocking the growth of F-actin. This ensures the efficient generation of protrusion force and depolymerization of filaments disconnected from the membrane. Indirect and direct steric regulators prevent CP binding [19].

Both the axon and dendritic shaft have regularly spaced actin rings beneath the neuronal membrane, separated by spectrin. This periodic structure, together with associated proteins, constitutes a structural and mechanical scaffold that strengthens the neuron. Beneath it are single fibrils and bundles of F-actin, oriented parallel to the long axis of the neural process [21,22]. Polarized actin filaments in the axon initial segment are preferentially oriented with their barbed end towards the cell body, regulating vesicle transport to dendrites and axons with the help of Myosin Va and Myosin VI, respectively [23]. G-actin is also present in cell nuclei, where it can polymerize, forming structures that look different from classical F-actin. Here, actin regulates the activity of RNA polymerases, transcription factors, chromatin remodeling complexes, and histone deacetylases, and ensures the movement and remodeling of chromatin [24].

In neurons, actin is predominantly localized in presynaptic axonal boutons and postsynaptic dendritic spines (Figure 1). Here, it regulates synaptic vesicle release and endocytosis, neuronal shape, receptor trafficking, and receptor positioning in postsynaptic membrane [25]. In the core region of the presynaptic bouton, actin serves as a scaffold for the reserve pool synaptic vesicles, which are docked to it by synapsin. Ca^2+^ influx through voltage-gated Ca^2+^-channels (VGCCs) lead to calmodulin-dependent kinase (CaMKII)-induced phosphorylation of synapsin and vesicle dissociation. The vesicle can then move to the active zone near the synaptic cleft, where a series of Ca^2+^-dependent processes occur, leading to the release of the neurotransmitter. Actin can also guide synaptic vesicles to the active zone with the help of myosin or serve a barrier for vesicles fusion [25,26]. Therefore, actin and actin-binding proteins should be closely linked to presynaptic neuroplasticity and some forms of learning. In fruit flies, loss of synapsins decreases heat avoidance, impairs olfactory STM, and accelerates fading of courtship memory [27].

Both excitatory and inhibitory presynapses contain three pools of F-actin: the actin mesh at the active zone, actin rails that connect the active zone to the deeper synaptic actin pool, and dense, branched actin corrals encasing synaptic vesicles. The actin mesh and actin corrals are Arp2/3 dependent, while actin rails are formin dependent. The actin mesh may be involved in the formation of exocytic sites, actin corals appear to organize a reserve pool of vesicles, and actin rails are probably involved in vesicle trafficking between them [28].

Actin is also the main cytoskeleton component of dendritic protrusions—filopodia and spines [29]. Filopodia are long, thin, do not have a bulbous head, and do not form synaptic contacts. They are highly unstable, being abundant in young animals, for example, in the dendrites of layer 5 pyramidal neurons in mice. In contrast, spines are more stable and make up the vast majority of dendritic protrusions in mature mice. About 73% of them last over 1.5 years, which is almost the entire lifespan [30]. Although dendritic spines are best described for excitatory synapses in mammals, they are also widespread in other vertebrates, as well as in some invertebrates, including fruit flies, squid, leeches, and flatworms [31]. Dendritic spines can be dually innervated by both excitatory and inhibitory presynaptic neurons: synaptic N-methyl-D-aspartate receptors (NMDAR) currents are reduced here through inhibitory tonic GABA_B_ receptors. The probable effect is spine stabilization [32]. Similarly to excitatory synapses, actin regulates activity of inhibitory synapses, although the effects of actin remodeling on synaptic activity appear to be somewhat different. For example, in excitatory synapses, actin polymerization and depolymerization are mainly associated with long-term plasticity and long-term depression (LTD), respectively. In contrast, in inhibitory synapses, both increases and decreases in inhibitory potential amplitudes may be associated with actin polymerization [33].

Spines usually have a head that is attached to the dendritic shaft by a neck. There are four morphological types of dendritic spines–mushroom spines, with a large head and a narrow neck, thin spines (less stable than mushroom spines), stubby spines, and branched spines with more than one heads. F-actin is a predominant form of actin in spines, forming either branched network or thick bundle of filaments cross-linked by specific proteins. The branched actin filament network is a dominant actin form of both dendritic spines and axonal boutons. The neck and base of the spine contain a mix of branched and linear actin filaments [34]. Dendritic spines also include cell membrane receptors, such as NMDAR and α-amino-3-hydroxy-5-methyl-4-isoxazolepropionic acid receptor (AMPAR), postsynaptic density (PSD) proteins, small GTPases, and other proteins participating in actin remodeling. Actin filaments are linked to the extracellular matrix via cell adhesion molecules [29]. Some mRNAs and microRNAs are also present in dendritic spines, such as miR-134, which regulates spines development [35].

Dendritic spines contain three pools of F-actin: 1. A relatively stable pool at the base of the spine, which appears to define the spine stability. 2. A dynamic pool at the tip of the spine generating the expansive force, which defines the size of the spine. 3. An enlargement pool distributed throughout the spine head, which is responsible for the activity-dependent growth of the spine. All pools are labile, with different turnover constants [36]. About 85% of actin pool has a turnover time of about 44 s. NMDAR-dependent influx of Ca^2+^ stabilizes about half of dynamic actin [37].

Together with the actin cytoskeleton, microtubules play an important role in the structural organization and functioning of synaptic terminals. In axons and dendrites, microtubules ensure neuronal polarity and control selective trafficking of cargoes via specific motor proteins [38]. Filopodia that emerge from the dendritic shaft contain branched actin and microtubules [34]. Ca^2+^ influx through NMDAR promotes microtubules to dendritic spines, which depends on local actin remodeling [39]. Thus, two major components of the neuronal cytoskeleton—actin filaments and microtubules—closely interact to govern spine morphogenesis and neuroplasticity.

## 3. Regulation of Actin Remodeling in Nervous Cells

Acting together, the actin-associated proteins regulate the complex process of formation, maintenance, and dynamics of the actin cytoskeleton. It would be impossible to give them all the attention in this review. Let us take a closer look at the proteins of the ADF/cofilin family, as their role in actin remodeling and memory processes has been best studied.

Actin depolymerizing factor (ADF) and cofilin are expressed in virtually all eukaryotes. Usually, they are located in parts of the cell with a high turnover of F-actin [40]. For example, in hippocampal dendritic spines, cofilin is predominantly localized at the periphery, near the cell membrane and PSD. Here, it appears to provide actin lability, while stable F-actin rests in the core of the spines [41]. Cofilin has both a nuclear export signal (NES) and a nuclear localization signal (NLS); its transport into the nucleus is increased by stress [42]. Cofilin also participates in the transportation of actin to the nucleus [43]. Cofilin-1 and cofilin-2 are the major forms in non-muscle and muscle tissue, respectively. They differ somewhat from ADF in their mode of regulation, susceptibility to oxidation, ability to nucleate and depolymerize actin, and survival requirements [44].

There are several ways in which ADF/cofilin proteins affect the actin cytoskeleton. Their mode of action also depends on their concentration. ADF/cofilins enhance treadmilling and increase the rate of ADP-actin dissociation [45]. Another actin-remodeling factor, profilin, directs G-actin to the barbed end of F-actin. When acting together, cofilin and profilin result in a 125-fold increase in the treadmilling rate [46,47]). Cofilins also sever actin filaments, most efficiently at a low ratio (<1:100), while at a high ratio they stabilize actin filaments. Higher concentrations of cofilin bind cooperatively to F-actin and stimulate its nucleation from G-actin [48,49]. ADF/cofilin also promotes P_i_ release from ADP-P_i_ filaments and actin debranching via dissociation of F-actin from Arp2/3 [50].

ADF/cofilin activity is primarily regulated by LIM kinases, a family of serine/threonine protein kinases, which includes LIMK1 [51] and LIMK2 [52]. Both of them contain two LIM (LIN-11, Isl-1, MEC-3) domains, a Dlg-homologous PDZ domain, a Ser/Pro-rich domain, and a C-terminal protein kinase domain. LIM and PDZ domains are involved in protein–protein interactions and regulate kinase activity [53]. The PDZ domain includes two NES. LIMK1 and LIMK2 also have a NLS, which allows them to perform some of their functions in the nucleus [54,55,56]. Proteins of the LIMK family have been found in humans, rats, chickens, frogs, and fruit flies. Drosophila has only one LIMK gene (*limk1*), encoding three long and two short protein isoforms [57]. The mammalian TESK, which is expressed in the testes, has an N-teminal protein kinase domain similar to that of LIMK [58]. LIMK1/TESK-dependent phosophorylation of Ser^3^ blocks the ability of cofilin to interact with G- and F-actin [44,59]. The phosphatases SSH and CIN reactivate ADF/cofilin [60,61], while SSH1 also inactivates LIMK1 [62]. Scaffold proteins β-arrestins provide interaction of LIMK1 with cofilin and CIN [63]. HSP90 stimulates dimerization and transphosphorylation of LIMK1, increasing its half-life [64].

In turn, LIMK1 is an effector of Rac, Rho, and Cdc42, which belong to the Rho family of small GTPases. The Rho–ROCK and Rac/Cdc42–PAK1 signaling pathway phosphorylates Thr^508^ in the LIMK1 activating domain [62,65,66,67]. Rac1 is an important regulator of various cellular processes, including stress reaction, remodeling of the cytoskeleton, dendritic spine morphogenesis, and memory [68,69]. Activation of LIMK1 by Rho–ROCK or Rac/Cdc42–PAK results in the phosphorylation of cofilin [70]. The non-canonical PAK-independent Sickie-involving Rac–cofilin pathway acts antagonistically to LIMK1 by activating SSH, which dephosphorylates cofilin [71]. Actin is enriched in Drosophila spines several-fold after Rac1 inductions, which likely depends on α7 nicotinic acetylcholine receptors [72]. In addition to LIMK1-dependent cofilin phosphorylation, Rac1 and Cdc42 also activate JNK and ERK signaling pathways, which induce gene transcription [73,74]. LIMK1 can also be activated by some other kinases, including PKA [75].

Rho family proteins act downstream of various cellular receptors. The BDNF-dependent TkrB receptor activates MECP2 and CDK1.5, which in turn activate Rac1 [68]. Rac1 activation by BDNF shifts CYFIP1 from the complex with the translation factor eIF4E into the WAVE regulatory complex. Both pathways induce actin polymerization and spine maturation [76]. EphB2 receptor-dependent phosphorylation of cofilin is mediated by FAK, ROCK, and LIMK1 [77]. The D1 receptor of dopamine activates Rac1 and Cdc42, while the D2 receptor inhibits Rac1 but activates Cdc42 in the nucleus accumbens [78]. In rat hippocampal neurons, BDNF relieves miR-134-dependent inhibition of LIMK1 translation in dendritic spines, increasing their size [35].

Ionotropic receptors of glutamate also influence Rho family proteins and remodeling of the actin cytoskeleton. NMDAR triggers activation of Rac1, stimulating its association with the NR1 receptor subunit [79]. Synaptic NMDAR-dependent stimulation of two signaling cascades, CaMKI–βPIX–Cdc42–PAK1–Shootin1a and CaMKII–Tiam1/Kalirin-7–Rac1–WAVE–Arp2/3, enhances actin polymerization [80]. In the nervous system, Rac and other Rho family proteins are activated by multiple guanine nucleotide exchange factors and inhibited by GTPase-activating proteins, which regulate different functions. Among them are Tiam1 (spine morphogenesis), Kalirin-7 (synaptic plasticity, learning, and memory), Kalirin-12 (dendritic outgrowth and branching), RasGRF1/2 (synaptic plasticity), and many others, which receive upstream signals from different cellular receptors and messengers [81]. Figure 2 shows the major signaling pathways that influence actin polymerization.

## 4. Involvement of the Actin Cytoskeleton in Synaptic Plasticity, Synaptogenesis, and Development

Synaptic plasticity can be defined as “the activity-dependent modification of the strength or efficacy of synaptic transmission at preexisting synapses” [82]. Short-lasting forms of synaptic plasticity, which occur over time intervals ranging from milliseconds to several minutes, are associated with transient presynaptic changes. The above include altered Ca^2+^ levels and activity of synaptic proteins, changing the probability of neurotransmitter release. Long-lasting changes in synaptic strength include long-term potentiation (LTP) and LTD. Each of them may develop through different mechanisms, such as increased/decreased neurotransmitter release, as well as insertion/internalization of postsynaptic AMPAR [83]. In turn, LTP can enhance synaptogenesis by altering the morphology of existing synapses and stimulating the formation of new synapses [84]. In the late phase of LTP, the formation of new mature spines is observed between the activated neuron and its target dendrite [85].

According to the model described in [86], the growth of dendritic spines is regulated by the following rules: 1. New actin polymerization foci are generated near the PSD. 2. The predominance of polymerization over depolymerization generates an expanding force that deforms the lipid membrane, which counteracts it. 3. The expanding force is proportional to the number of non-capped barbed ends that can branch from actin filaments. 4. The barbed ends can be capped or uncapped, which decreases or increases the expanding force, respectively. The uncapped filaments can also be severed. 5. The probability of branching decreases with increasing number of barbed ends. The latter creates a negative feedback, allowing the system to respond more rapidly to external signals. The model predicts the experimentally observed fluctuations of spine shape. Thus, actin dynamics are crucial for synaptic plasticity.

In cells, nucleation, assembly, and disassembly of actin filaments are governed by specific signaling factors. Rac1 is necessary for LTP induction in the hippocampal neurons [87]. In rat hippocampal dendritic spines, tetanic stimulation shifts the F-actin/G-actin equilibrium toward F-actin, increasing their size and postsynaptic binding capacity. Low-frequency stimulation produces the opposite effect [88]. The early phase of synaptic potentiation is caused by AMPAR- and NMDAR-mediated actin polymerization and enlargement of dendritic spine heads. Long-lasting spine enlargement requires CaMKII [89]. RhoA and Cdc42 are also activated in stimulated spines. Rho–ROCK signaling pathway regulates spine growth, while the Cdc42–PAK pathway is required for sustained plasticity. Both RhoA and Cdc42 are activated by CaMKII [90]. Rac3 is more efficient in promoting spine enlargement compared to Rac1 [91].

Rac1 is required for LTP induction and is inhibited during the LTP maintenance phase. In the former case, Rac1 acts through the PI3K–PKC_*ι*/*λ*_ pathway. In the latter case, the Rac1–LIMK1 pathway inhibits PKMζ expression and suppresses LTP [92]. Since LIMK1 can phosphorylate CREB, influencing gene transcription [93], its effect on LTP maintenance is likely actin-independent. Cdc42 activity is required for LTP induction and time-restricted maintenance, but here it acts through cofilin phosphorylation. Cdc42 also induces heterosynaptic cooperation and competition between spines closely spaced on the dendritic tree. Actin remodeling is required for both synaptic cooperation and competition [94]. Thus, complex, non-linear forms of synaptic interaction are observed, where Cdc42-dependent phosphorylation of cofilin does not actually lead to F-actin stabilization, but enhances its dynamics.

After LTP induction, stable F-actin in dendritic spines is disassembled by unbundling and severing by cofilin. This facilitates the next phase, when actin rapidly polymerizes and branches with the help of Arp2/3, increasing the spine volume. Finally, the new actin filaments are bundled and crosslinked [95]. Interestingly, CaMKIIβ can act as an actin-crosslinking protein, slowing the actin turnover in spine heads and maintaining the structure of mature spines. This function of CaMKIIβ does not depend on its kinase activity and occurs when it is inactive [96]. Thus, like cofilin, CaMKII is a multifunctional protein whose influence on synaptic neuroplasticity appears to be complex and non-linear.

During LTP induction, cofilin completely fills the head of the spine; its concentration increases rapidly and remains enriched for a long time. In contrast, actin and Arp2/3 concentrations rapidly stabilize, remaining stable during LTP and spine enlargement. They appear to be incorporated into the growing cytoskeleton as its major components [97]. The effect of cofilin on LTP is complex and involves several consecutively acting molecular mechanisms. Increased ADF/cofilin activity enhances actin dynamics, stimulating actin-dependent AMPAR trafficking and exposure on the spine surface. The latter enhances LTP without morphological changes in dendritic spines that lag behind AMPAR insertion into the membrane. Subsequent phosphorylation of cofilin is required for actin polymerization and spine growth [98]. Thus, active cofilin ensures actin lability, which is crucial for synaptic remodeling, while stable, branched forms of actin are required for the maintenance of remodeled synaptic connections. In the mouse hippocampus, the level of active ADF/cofilin dramatically increases during the synaptogenesis period. Active cofilin is required for AMPAR mobility; in postsynaptic terminals, cofilin regulates late LTP and LTD [99,100].

Other actin-remodeling proteins also influence synaptic plasticity. NMDAR-dependent translocation of profilin to dendritic spines stabilizes their morphology [101]. Profilin-1 triggers synaptogenesis early in development by stimulating actin polymerization. In contrast, profilin-2 is required for spine stabilization and activity-induced remodeling of spine structure in mature neurons. It slows down actin polymerization, presumably competing with other actin-binding proteins such as Arp2/3 [102]. Thus, like cofilin, profilin or its specific types exert a dual effect on actin remodeling, which may be crucial for synaptic plasticity and adaptation of synaptic patterns to learned cues.

In fruit flies, synaptic plasticity is regulated by multiple signaling factors, including Wnt, FasII, CaMKII, PKA, and CREB [103]. Wingless/Wnt regulates synaptic plasticity at the Drosophila neuromuscular junction (NMJ) via presynaptic cortactin, a membrane protrusion regulator, which activates Arp2/3-dependent actin branching and induces activity-dependent synaptic modifications [104,105]. In the NMJ, BMP type II receptor Wit promotes formation of presynaptic boutons through LIMK/p-cofilin-dependent actin polymerization [106].

Actin-remodeling factors are also important for the development of the nervous system. Loss of Rac GTPases leads to defects in Drosophila axon branching, guidance, and growth [107]. Both active and phosphorylated forms of cofilin are required for axonal growth in the fruit fly mushroom bodies (MB), suggesting cycles of cofilin inactivation by LIMK1 and reactivation by SSH. Both of the above processes are Rac-dependent [70]. The axonal growth depends on Sickie, which activates SSH. In the *sickie* mutant strain, the F-actin signal is greatly increased around the lobe branching point of the MB axons. Since actin is limited in neurons, cofilin appears to stimulate axonal growth by increasing F-actin turnover, providing the growth cone with actin monomers [71]. This can also explain the stimulating effect of cofilin on spine growth, being in accordance with the model of disassembly/assembly of F-actin during LTP [86,95].

The motility of the axon growth cone is determined by three main factors: 1. Actin polymerization, which generates tensile forces that bulge the cell membrane and create a return movement of F-actin bundles. 2. Myosin-dependent contractile forces involved in F-actin retrograde flow. 3. Clutching forces arising from the interaction of the F-actin cytoskeleton with the extracellular substrate through specific transmembrane and adapter proteins. All these interactions are dynamically regulated by multiple chemoreceptors, mechanosensory receptors, and signaling proteins, providing the necessary speed and direction of growth cone movement. ADF/cofilin-dependent depolymerization at the F-actin minus end decreases compressive forces of the actin network and promotes polymerization at the plus end [108].

In rat hippocampal neurons, maturation of the mushroom spines requires contractile activity that depends on actin and non-muscle myosin IIB (MIIB). Contractile activity is stimulated by NMDAR, which induces ROCK-dependent diphosphorylation of MIIB regulatory light chain [109]. In Drosophila NMJ, myosin II is involved in the formation of presynaptic cable-like actin structure that senses axonal mechanical tension and presumably transduces mechanical signals to muscles via integrin-β [110].

## 5. The Role of Actin Cytoskeleton in Learning and Memory Retention

Although LTP and synaptic plasticity have long been considered the basis of memory, experimental evidence of this has only recently been obtained using memory engram technology [111,112]. Functional activation and optogenetic stimulation identified engram cells that promote encoding, consolidation, and recall of a particular memory. Engram cells are active both during initial learning and during memory retrieval. They also exhibit persistent changes in their internal properties and connections with other neurons that serve as the physical basis of memory [113]. The above changes include selective synaptic potentiation, as well as an increase in spine volume and head diameter [114]. During learning and novel sensory experience, new spines are formed, while some existing spines are eliminated. The degree of spine remodeling is proportional to the behavioral changes after learning. A small part of novel spines persists throughout life, possibly serving as the basis for long-term memory (LTM) [115]. Specific memory engrams can be associated with different synaptic connections of the same neuron [116]. The key link in this process appears to be the actin cytoskeleton and the signaling cascades that regulate its remodeling.

In mammals, acquisition of some types of memory requires activation of Rac1, which should result in cofilin inhibition, F-actin stabilization, and spine maturation. Most studies were conducted on mice. Activation of cofilin in the hippocampal CA1 region of mice disrupts 24 h object-place LTM, reducing the length and density of dendritic spines [117]. Rac1 also regulates hippocampus-dependent spatial learning and episodic-like memory, probably acting through PAK and LIMK. Mice lacking Rac1 have fewer number of synapses [87]. Both STM and LTM of fear conditioning require activation of Rac1 in the basolateral amygdala (BLA), although the signaling pathways downstream of Rac1 involved in memory formation are unknown [118]. Inhibition of presynaptic Rac1 in mouse hippocampal neurons impairs spatial working memory, affecting the distribution and morphology of synaptic vesicles. Presynaptic Rac1 interacts with many partner proteins, including Wasf3 and Abi2, which are involved in the organization of actin filaments. Inhibition of postsynaptic Rac1 impairs contextual fear LTM [119]. Rac1 also stimulates reconsolidation of fear memory in the rat BLA [120]. It should be noted, however, that cofilin activity was not specifically investigated in most of the above studies.

In contrast, other forms of learning and memory require active cofilin. NMDAR-dependent translocation of cofilin to hippocampal dendritic spines is necessary for LTD and spatial LTM, but not STM [121]. In the mouse forebrain, cofilin deficiency impairs AMPAR mobility and associative learning, but does not influence exploratory learning and short-term working memory [100]. Hyperactivation of cofilin due to blockade of LIMK improves hippocampus-dependent STM [122].

Such contradictory effects of cofilin on memory processes can probably be explained similarly to its dual effects on LTP. The formation of new memory initially requires active cofilin and an increase in actin turnover, which allows the actin cytoskeleton to restructure and provides material for new actin fibrils. Then comes the stabilization phase, when cofilin activity must be blocked. Both processes may partially overlap in time and compete. The situation is complicated by the involvement of F-actin in various cellular processes, such as receptor anchoring, protein transport, regulation of neurotransmitter release, and others. As a result, the formation of different types of memory in different brain areas or neuronal structures will correspond to a specific temporal pattern of cofilin activation and inactivation.

Spinogenesis requires proper assembly of actin with sufficient branching. Therefore, factors regulating actin branching should play an important role in memory processes. Among them is WAVE, which is primarily localized in the spine heads. In the mouse hippocampus, WAVE-1 loss in mature neurons impairs synaptic plasticity. WAVE-associated GTPase-activating protein WRAP is necessary for spatial and nonspatial memory [123].

Other factors influencing actin remodeling and memory processes include PKA and profilin. It is well established that PKA regulates the formation of STM through phosphorylation of synaptic proteins and LTM through the MAP kinase cascade and CREB [124]. However, in mice, cAMP/PKA also induces sleep-dependent memory consolidation through LIMK1 and cofilin. Sleep deprivation causes PDE4-dependent cAMP degradation and cofilin activation, leading to spine loss and LTM impairment [117]. In the rat lateral amygdala, profilin in complex with VASP and Arp2/3 stimulates the formation of fear LTM by stabilizing the actin cytoskeleton of dendrites. The above confirms the role of branched actin in memory consolidation [125].

Actin capping also regulates memory formation. In *Caenorhabditis elegans*, ADD-1, an α-adducin ortholog, stimulates olfactory STM and LTM by stabilizing F-actin, likely through capping of its barbed end. In humans, some *ADD-1* genetic variants are associated with episodic memory performance [126]. In cultured hippocampal neurons, CP participates in actin-binding-dependent stabilization of dendritic spines [127] and may therefore be involved in memory processes. However, the role of actin capping proteins in learning and memory remains poorly understood.

A challenging question is how the labile structure of the actin cytoskeleton in dendritic spines can provide spine-specific long-term storage of memory. A model was proposed to explain the above paradox: 1. Large spines are more likely to activate spontaneously, which in turn reduces their actin dynamics and preserves them. 2. The balance between actin polymerization/depolymerization and capping/decapping proteins maintains a stable structure of the actin cytoskeleton. 3. Actin remodeling proteins are actively translocated into or translated in specific dendritic spines [128]. The latter corresponds to the concept of synaptic tags: activation of certain synapses creates some molecular markers that attract newly synthesized proteins and/or mRNAs [129,130]. A stable, cross-linked pool of F-actin, which persists for up to 2 h after LTP induction, can serve as such a tag [131].

Proteins and signal cascades that influence actin cytoskeleton remodeling are involved in multiple forms of Drosophila behavior, such as olfactory classical conditioning, courtship memory, sleep, and reaction to ethanol [132]. Some actin- and microtubule-associated proteins, such as myosin 15, regulate presynaptic homeostasis plasticity in the NMJ, as well as middle-term olfactory memory in the MB [133]. Moesin, a cytoskeletal adapter protein that links actin to the cell membrane [134] and regulates axonal and spine-like protrusion growth during Drosophila development, also stimulates courtship memory consolidation in adult flies [135]. Moesin is activated by Rho-kinase [136], but itself acts antagonistically to the Rho pathway [137]. At the same time, no stimulating effect of Rho family proteins or LIMK1 on memory formation was detected in Drosophila. In contrast, Rac1 and other Rho-like proteins specifically regulate forgetting in the fruit fly [138], as discussed in the Section 6.

## 6. Actin Cytoskeleton and Memory Forgetting

When studying active forgetting, it is important to carefully distinguish between true forgetting and impaired memory formation. In the first case, memory typically extends beyond its normal lifetime. To prove it, it is necessary to suppress the mechanisms that normally activate forgetting, which has not always been the case [139]. However, this was indeed performed in the study by Shuai and colleagues, who discovered a signaling pathway that activates intrinsic forgetting. Hyperactivation of the Rac1–PAK–cofilin pathway in αβ and γ neurons of the Drosophila MB induces forgetting of olfactory STM, whereas inhibition of Rac1 prolongs protein synthesis-independent STM for up to 24 h [140].

Forgetting different types of memory involves different members of the Rho family and neurons of the MB. The Rac1–SCAR/WAVE–Dia signaling pathway in γ neurons activates forgetting of anesthesia-sensitive memory (ASM), while the Cdc42–WASP–Arp2/3 pathway activates forgetting of anesthesia-resistant memory (ARM) [141,142]. Dia is homologous to formin, which facilitates polymerization of linear actin [143]. The involvement of Arp2/3 in the latter case may be explained by the fact that ARM is more robust and likely depends on mature spines, which contain Arp2/3 [144]. N-WASP stimulates the ability of Arp2/3 to nucleate actin polymerization [145]. Arp2/3 also regulates forgetting in *C. elegans.* MSI-1-dependent repression of *acx-1*, *arx-2*, and *axr-3* mRNAs translation decreases Arp2/3 activity and stimulates memory loss, probably via inhibiting actin branching [146]. Both actin polymerization and depolymerization factors appear to be involved in forgetting [143].

In fruit flies, forgetting is regulated by specific dopaminergic neurons and dopamine receptors. While memory formation depends on the Dop1R1–G_S_–cAMP signaling pathway, memory erasure is caused by the Dop1R2–G_αq_–Ca^2+^ signaling pathway [147,148,149]. Both Dop1R1 and Dop1R2 are homologous to D1-like mammalian receptors, of which only Dop1R2 is capable of activating G_αq_ [150]. Dop1R1-dependent olfactory learning drives *rutabaga*-encoded adenylate cyclase and PKA [151]. In turn, interference-based forgetting is activated by the Dop1R2–Rac1–PAK3–cofilin pathway in the αα’ MB neurons [152]. Intrinsic and interference-based forgetting require Rac activity in certain MB output neurons [153]. How the structure of the actin cytoskeleton changes during the processes described above remains to be determined.

Immediate courtship memory is also impaired in mutant *agn^ts3^* with increased LIMK1 and p-cofilin levels [154,155]. Activation of *limk1* in the MB with concomitant increase in LIMK1/p-cofilin levels impairs 30 min courtship memory compared to control and knockdown flies [156]. The above highlights the role of actin polymerization in forgetting, which can occur so quickly that the memory is effectively absent from the start. Butts and colleagues showed that increased cofilin activity and F-actin dynamics in the Drosophila MB enhance the acquisition of alcohol consumption preference, whereas Rac activation and F-actin stabilization suppress it. The authors view this as a type of forgetting, in which information is forgotten right away after learning. At the same time, F-actin stabilization may be required during memory consolidation or recall [157].

Rac-dependent forgetting has also been shown for mammals. In mice, stimulation of postsynaptic Rac induces shrinkage of the dendritic spines in the motor cortex and disruption of motor learning [158]. Rac1 and PAK inhibit auditory fear memory [159]. The temporal profile of Rac1 activity determines whether memory will be stored or forgotten. In the hippocampus of adult mice, loss of Rac1 impairs learning-evoked neurogenesis and causes prolonged retention of spatial memory [160]. The latter is an example of neurogenesis-based forgetting activated by new learning. In the study of Liu and co-authors [161], activating or inhibiting Rac1 in the mice hippocampus did not affect their learning ability and object recognition memory, but impaired or prolonged 72 h memory, respectively. The expression of dominant-negative Rac1 resulted in the formation of thinner, filopodia-like spines, whereas Rac1 hyperactivation resulted in the formation of spines with larger heads. Rac1 activation also induced interference-based forgetting and accelerated LTP decay. No changes in contextual fear conditioning were observed. Hence, Rac1 seems to act differently in specific brain regions or at different time points: its activity during learning is necessary for memory formation, but after a certain period of time causes forgetting.

As in mammals, synaptic plasticity and memory consolidation in the fruit fly must involve remodeling and stabilization of dendritic spines and, consequently, actin polymerization. Little is known about the fine structure of the Drosophila spines and their learning-dependent changes. A dynamic balance of active and phosphorylated cofilin is required for axonal growth [70], likely because active cofilin stimulates F/G-actin turnover [71]. Dynamic disassembly and reassembly of F-actin is required for dendritic spine growth after LTP induction [86,95]. Thus, increased Rac1 activity should lead to a shift in the balance towards F-actin, disrupting synaptic plasticity, which, in turn, should impair learning of new information.

It is somewhat surprising, however, that Rac1 is not required for memory acquisition in Drosophila [140]. To the best of our knowledge, there is no experimental evidence for the involvement of Rac or LIMK1 in fruit fly olfactory memory formation. At the same time, our recent studies have shown that both activation and suppression of *limk1* in fruit fly cholinergic neurons impair short-term courtship memory, although in the former case, the memory decays more rapidly [156]. Thus, *limk1* activity in some as-yet-unidentified parts of the Drosophila brain may be necessary for maintaining courtship memory.

In mammals, Rac1-dependent and Rac1-independent memory formation pathways converge at the level of PKA [117,124]. It is not clear whether PKA can activate LIMK1 in Drosophila. In the fruit fly, the pathways regulating memory formation and forgetting may be more strictly separated. Actin polymerization in Drosophila dendritic spines may be primarily regulated by some adapter proteins, such as moesin [135]. The Rac-dependent pathways also compete at the level of LIMK1 and SSH, which have opposing effects on cofilin activity [71]. The final effect appears to be determined by the magnitude or timing of Rac1 activation. Such antagonistic signaling interactions are crucial for memory formation, which requires spaced training. In rats, spaced but not massed training induces inhibition of Rac1 in the hippocampus, enhancing contextual fear. Rac1 activity was suppressed 1 h after spaced training, but remained unchanged after massed training [162]. Similarly, repetition suppresses the activation of Cdc42, which causes ARM forgetting [141]. Thus, the competition between memory and forgetting processes depends on the orchestrated activity of many signaling proteins, which may differ somewhat between Drosophila and mammals.

It is unclear how exactly actin polymerization and branching induce intrinsic forgetting. There seem to be two possible ways: 1. Interference with actin turnover, which is necessary for synaptic plasticity and the formation of new memories. 2. Non-specific formation of new spines, the activity of which interferes with the activity of memory-induces spines. Spontaneous synaptic changes occur in the brain and challenge the existing memories [163]. Background Rac1 activity may influence such spontaneous spine dynamics and induce forgetting through non-specific stabilization of some spines and erosion of specifically formed synapses, responsible for memory traces. Memory and forgetting may involve the activity of different neurons, although at least in some cases they appear to occur in the same neuronal population [164]. Competition for free G-actin may occur at the level of different synaptic spines. The contradictory effects of Cdc42-dependent cofilin phosphorylation and actin depolymerization were observed during heterosynaptic interactions [94]. Thus, the ultimate effect of a particular actin-remodeling factor is hard to predict because it depends on the molecular and cellular context.

In Drosophila, mutation of *dfmr1* encoding the fragile X messenger ribonucleoprotein increases the level of profilin [165] and impairs immediate courtship memory [166]. Thereby, in fruit flies, profilin appears to act similarly to p-cofilin, causing memory loss. Both proteins stimulate actin polymerization. However, while profilin actively promotes F-actin formation, p-cofilin passively promotes it after being inactivated by LIMK1, or in other words, after it has played some role in F-actin disassembly. Since learning naturally precedes forgetting, there must be some time delay between the action of cofilin as a permitter of new memory formation and (in its inactivated form) as a repressor of actin remodeling, which stabilizes the existing structure of the cytoskeleton. Returning to our analogy of fusing neurons, cofilin acts as a “solder” that first, in its “hot”, active form, allows synaptic spines to “melt”, and then fixes their new shape and contacts. Profilin and Arp2/3 appear to act in the final step to perpetuate cytoskeletal changes.

Forgetting also engages the activation of specific genes. In some Drosophila dopaminergic neurons innervating the MB, NO serves as a cotransmitter, which alters the modality of olfactory memory [167]. In α/βs and γm neurons of the MB, the NO-dependent signaling cascade stimulates the histone H3K9 demethylase, Kdm4B, and the GMP synthetase, Bur, which in turn induce the expression of *kek2.* The above leads to forgetting, since *kek2* suppression enhances 6 h memory. This memory decay has been called “gene expression–based forgetting”. *kek2* potentially encodes a receptor for neurotrophins, such as BDNF. Although the cognate cellular receptor can activate Rac1, the above processes are independent of Rac and Raf [168]. The related gene *kek6* regulates synaptic plasticity via CaMKII [169]. It would be interesting to investigate how the structure of the actin cytoskeleton changes during these processes.

Another mechanism that maintains the balance between memory and forgetting is glia-dependent phagocytosis of neuronal synapses. In the mouse hippocampus, astrocytes specifically eliminate excitatory synapses in the adult CA1 region. Although mice with defects in phagocytosis have a larger number of excitatory spines, they are also characterized by reduced presynaptic neurotransmitter release, decreased LTP and LTD, and memory impairments [170]. The above shows the need for a precise balance between excitatory and inhibitory signals to ensure the optimal conditions for synaptic plasticity and memorization. Microglia participate in forgetting after contextual fear conditioning by selectively phagocytosing synaptic components of hippocampal engram cells with decreased activity [171]. The above is performed in accordance with Hebb’s principle, since weak and non-functional connections should be eliminated. Astrocytic Rac1 also appears to induce forgetting. In mice, fear memory suppresses Rac1 activity in BLA and reduces the volume of astrocytes. Rac1 ablation promotes memory formation, whereas Rac1 activation attenuates learning in a p-cofilin-dependent manner [172]. Thus, Rac1 may cause memory erasure indirectly by acting on the glial actin cytoskeleton. The effects of actin-remodeling proteins on memory and forgetting are summarized in Table 1.

Stimulation of neurogenesis in the hippocampus induces forgetting in rodents after contextual fear conditioning, incidental context learning, and the Morris water maze training. The cellular basis of forgetting appears to be the formation of new dentate gyrus–hippocampal CA3 synaptic connections near existing ones, which disrupt their activity [173]. Since Rac-dependent signaling pathways are involved in neurogenesis, they likely play a role in interference-based forgetting, as shown in [152].

## 7. Actin Remodeling and Neuropathologies

Abnormal cofilin activity and actin remodeling lead to the disruption of neural connections and engrams in many neuropathologies. These include Alzheimer’s disease (AD), Parkinson’s disease, Huntington’s disease, stroke, schizophrenia, amyotrophic lateral sclerosis, and others. Increased cofilin activity induced by β-amyloid, hyperphosphorylated tau, and ischemic stroke leads to decreased actin stability, resulting in loss of dendritic spines and axonal damage [174].

Synaptic degeneration is one of the most typical features of early AD. A stable actin cytoskeleton may protect synaptic spines from elimination, serving as a kind of “synaptic safety net” [175]. Both activation and inhibition of the LIMK1-cofilin-actin pathway are involved in AD neuropathology at different stages, highlighting the dual role of cofilin in the disease progression [176]. In cultured primary rat hippocampal neurons, Aβ42-dependent activation of ROCK2 and LIMK1 causes spine loss and hyperexcitability, which are blocked by a LIMK1 inhibitor [177]. Spine loss is reminiscent of what occurs in LIMK1-dependent forgetting, which may also contribute to cognitive defects in AD. At the same time, induced pluripotent stem cells derived from patients with sporadic AD show a decrease in phosphorylated LIMK1 and cofilin, as well as a decrease in the length and complexity of dendrites [178]. In this case, LIMK1-dependent actin polymerization, which is necessary for proper dendritic development, appears to be impaired. Thus, we are again faced with the dual effects of LIMK1-dependent signaling, the deviation of which from the optimum in one direction or another can lead to neurological defects.

Activity of Rac/Cdc42–PAK signaling pathway is reduced in the cortex of AD mice. Meanwhile, levels of inactive pRac/Cdc42 are elevated in the cortex of the human AD brain. Spatial and subregional differences in pRac/Cdc42 levels, as well as in the expression of several genes associated with Rac/Cdc42 signaling, are observed. Their functional effects are unclear. However, they point to specific features of Rho-dependent signaling in different brain regions, both in normal conditions and in AD [179].

Little is known about the association between cognitive abilities and single-nucleotide polymorphisms (SNPs) in genes regulating actin remodeling. However, a specific SNP in the protomer of the *LASP1* gene is related to category verbal fluency and non-verbal working memory during schizophrenia. LASP1 is an actin-associated protein that has LIM and SH3 domains and resides in the PSD of dendritic spines [180]. In humans, *limk1* hemizygosity results in Williams syndrome, which is characterized by impairment in some forms of learning and visuospatial cognition [181,182].

A few small-molecular compounds can increase dendritic spine density, improving learning and memory in AD models. Among them are benzothiazole derivatives, which increase the association of fascin-1 with actin [183]. Genome-wide association studies of general cognitive abilities can help identify potential genetic targets for nootropic drugs that are already used as therapeutic agents in the treatment of certain diseases. Such putative targets include sets of genes encoding actin- and chromatin-binding proteins. Among the most likely sixteen target proteins are PDE4C (needs to be suppressed) and PDE4D (needs to be activated) [184]. PDE4A5 is known to impair sleep- and LIMK1/cofilin-dependent memory consolidation [117]. Thus, the actin cytoskeleton and factors regulating actin remodeling are promising targets for the nootropics and therapeutic agents against AD.

Aging has been shown to result in decreased actin polymerization and β-actin expression, as well as disruption in the integrity and morphology of the actin cytoskeleton, which ultimately leads to the impairment of multiple actin-dependent functions. This makes actin a possible target for anti-aging drugs [185,186]. In the mouse hippocampus, the p75 neurotrophin receptor, which is required for LTD, is responsible for the age-related decline of long-term synaptic plasticity and memory. It also causes an age-related decrease in the phosphorylation of LIMK1 and cofilin, possibly affecting the integrity of the actin cytoskeleton [187]. Thus, some of the negative effects of aging on cognitive performance are mediated by the activation of a specific signaling pathway that may serve as a target for therapeutic agents.

Under stress, activated cofilin saturates actin filaments, which form rod structures in the cytoplasm or nucleus and stop actin polymerization. These rods are involved in the progression of neurodegenerative diseases, probably impairing nuclear processes in the cell [188]. In aged Drosophila, some parts of the optic lobes show increased levels of F-actin and F-actin-rich rods, which disrupt brain autophagy and mitophagy, reducing locomotor activity and climbing index. Overexpression of ADF/cofilin, inhibition of the F-actin nucleator Fhos, and use of actin depolarizing drugs restore autophagy and prolong lifespan [189]. The above underscores the importance of a balance of cofilin activity and the F/G-actin ratio for maintaining neurological functions. Age-related impairment of LTM has been shown for the wild-type Drosophila strain *Canton-S* and the mutant strain *cinnabar*^1^ [190]. Although the mechanisms remain unclear, they may be related to defects in actin remodeling.

## 8. Natural Intelligence and ANNs

Studies of memory mechanisms at both the neural and molecular levels show that the brain is an extremely complex system with multiple levels of organization, extensive parallelism, and feedback loops. The holographic theory states that when the brain learns something, the engram is not strictly localized in any one part of it, but is dynamically distributed throughout the brain network [191]. Similar principles underlie the architecture and training of ANN: its connection pattern is adapted as a whole to solve a specific task, without any explicit rules for each neuron to interact with any other.

It is always tempting and at the same time dangerous to draw parallels between phenomena belonging to different scientific disciplines, which exhibit similarities in their behavior or principles of organization. The revolution in the field of ANNs that we are witnessing right now, and their undoubted success in solving problems traditionally considered the preserve of human intelligence, such as understanding and creating texts, makes us wonder whether the human brain is organized according to the same principles. The creators of ANNs were inspired by neurological discoveries, starting with the very idea of the brain as a complex system of neural connections, which specifically adapt to external stimuli. The technology of computer vision has made tremendous progress thanks to the architecture of deep convolution networks, whose basic principles are similar to those used in human object recognition [192]. The concept of attention, or selective detection of connections between the features of the analyzed objects, underlies the architecture of the transformer [193], which is the core of modern large language models. The paradox is that we now seem to understand better how ANNs work, using hundreds of billions of operations per second, gigawatt-hours of electricity demand, and billions of training samples to extract their general features, than how the human brain achieves the same thing at a much lower cost and often more efficiently.

Today, computer sciences can help us better understand how information is processed by the brain and even model it in silico as a special type of neural network (for example, see [194]). However, we must always remember that we may fall victim to false analogy, trying to attribute to a living system something that works well in theory, but is questionable or implausible from a biological point of view. For example, ANNs are typically trained using backpropagation, which iteratively adjusts the weights of artificial synaptic connections to minimize the error between the predicted and true outputs [195]. However, it is unclear whether such a mechanism of learning is possible for the brain. First, what is considered a “true output” in the case of the brain learning? The Bayesian brain theory proposes a somewhat similar model of how the brain might learn: it uses feedback between higher and lower regions to minimize surprise, or discrepancy between the predicted model of the world and the sensory stimuli used to create that model. Minimizing surprise is equivalent to minimizing the free energy of the system [196]. If we take surprise as the error measure, calculated at each level of the brain network, then its step-by-step minimization will resemble the process of iterative ANN training.

At the same time, the brain must have the ability to remember some complex information at once. This is especially important for episodic memory, which can form after a single presentation of an event, without repeated training. The auto-associative network proposed by Edmund Rolls appears to be better suited for this purpose, as it can learn very quickly and without backpropagation of errors. The only thing required for this is a feedback system that each neuron in the network forms with other neurons, strengthening their connections as they work in a coordinated manner [197].

When learning a given stimulus, the problem may not be its immediate memorization, but rather distinguishing it from other accompanying stimuli, or maximizing the mutual information of conditioned and unconditioned stimuli [198]. This can explain the fact that in many cases, one training session is enough for memory formation. Forgetting serves as a kind of defense mechanism that prevents remembering unstable, non-specific associations. Both passive and intrinsic active forgetting can be involved in this process. It is similar to the procedure of dropout in ANN, which randomly drops neurons and their connections during training to prevent overfitting [199].

It is well known that non-linear activation functions are needed for an ANN to learn associations between the features of objects. As for neural interactions in the brain, such non-linearity is ensured at several levels: 1. Release of neurotransmitter quanta. 2. All-or-none behavior of voltage-gated ion channels. 3. Competition between signaling cascades, such as Rac1-dependent pathways that are activated during learning and share some common components, but lead either to the formation of memory or to its forgetting. 4. Dual functional effects of some actin-binding proteins, such as cofilin, profilin, and CaMKII. 5. The complex three-dimensional network of actin cytoskeleton, which serves as a scaffold for synaptic terminals, regulates anchoring of vesicles and cellular receptors, while simultaneously blocking activity-dependent neuronal remodeling and traffic processes. At each level, parameters of interactions can be modified by learning and stored as forms of STM or LTM. The above creates an additional depth and flexibility in the brain compared to deep ANNs.

The “prospective configurations” model, suggested by Song and co-authors [200] instead of the backpropagation model, follows the concept of [196] and other energy-based networks. Here, the network is able to adapt its activity before adapting the weights of connections between neurons, converging to some energy minimum thanks to a feedback system. Such ANNs appear to learn faster, better reproduce patterns of brain activity, and perform better at solving many problems encountered by living organisms, such as contextual learning and reinforcement learning. Requiring no complex computations, they behave more like analog than digital devices.

Figure 3 compares the structures of natural and artificial neural networks. ANN is represented here by a fully connected (dense) network with backpropagation training. Each neuron in each layer is connected to each neuron in the next layer (shown by blue arrows), without intra-layer, across-layer, or feedback connections. An exception is the residual connection, where two neurons from the hidden layers 1 and 4 interact directly. Backward connections are simply the inverse of forward connections, transmitting an error signal (backpropagation). Such “naive” architecture does not know anything about the optimal topology of connections, which must be found iteratively. All synaptic connections are equivalent before training. The gradient of error is computed for each connection, and synaptic weights are updated step by step. Dropout is applied randomly. Such ANN architecture serves as a prototype for others, where the initial topology of connections is pre-designed to make it more suitable for solving specific tasks, such as computer vision. The natural neural network is presented in a simplified form, to make the differences from ANN more pronounced: 1. Neurons are colored differently to highlight their chemical and morphological diversity. 2. The pattern of connections is extremely diverse, with a large number of feedback, collaterals, and across-layer connections. 3. The principle of topical organization is fulfilled: neighboring neurons in one layer project primarily onto neighboring neurons in another layer. 4. Some parts of the brain are isolated from each other, performing different functions, and the connections between them are relatively few. 5. The number of deep layers is not as big as in modern ANNs, where it can reach thousands of layers. However, each brain layer can be represented by many millions of neurons, which greatly increases parallelism. 6. Each neuron itself represents a network of non-linear molecular interactions, which can be tuned. 7. Learning occurs according to the Bayesian brain model, where the difference between hypothesis and sensory stimuli is minimized.

The main advantage of the brain over dense ANN appears to be its developed system of feedbacks, both at the neuronal and molecular level, as well as specific parameters of topology, signaling cascades, and gene expression, selected by evolution for fast and relatively inexpensive learning. The latter may occur according to the principles described in [196,197,200]. It is well known that synaptic plasticity in the brain develop with a delay, in which consolidation of new synaptic connections can occur hours after training. The actin cytoskeleton in synaptic terminals plays a crucial role in this processes. The split between STM and LTM may reflect the separation of the weights of connection that adapt quickly and slowly after learning. Hence, the model described in [200] can be further refined to more accurately reflect brain function.

As with ANN, where it is impossible to predict how learning will affect the pattern and strength of neural connections, it is hard to predict how a specific signal protein or neural connection will affect engram storage. This apparently explains the discrepancies in the effects of cofilin obtained for different types of memory in different model objects. Similarly, HSAM may not be associated with any specific gene, but with an ideal balance of the activity of multiple genes and signaling factors, where the functions of each is hard to be interpreted outside the context of the others. This may explain why HSAM cases are so rare. A qualitative increase in the brain’s ability to remember and store information may be achieved not by interfering with its molecular processes, but by specially ordering the input data, taking into account its internal structure and the way how the brain processes it. The mnemonic techniques give examples of such data pre-processing.

## 9. Conclusions

There are still many questions regarding learning, memory, and forgetting, as well as the role of actin in these processes. What is the biological nature of HSAM? Is it determined by specific genes or a combination of multiple genetic alleles? Is it possible to turn off forgetting, for example, by suppressing some signaling proteins or interfering with the process of assessing the significance of new information? Or, conversely, how can one induce a deliberate forgetting of a specific traumatic memory? Does forgetting provoke a targeted destruction of specific spines or, rather, the non-specific formation of spines and/or F-actin fibrils that compete for G-actin and actin-binding proteins? What is the role of actin capping proteins in memory and forgetting processes?

Although the first genes regulating behavior and learning were discovered in fruit flies [201,202], much less is known about the processes of synaptic plasticity and learning-dependent actin remodeling in Drosophila. Which signaling pathways primarily regulate actin polymerization in the fruit flies dendritic spines upon memory formation? Forgetting in Drosophila has been studied primarily using associative olfactory learning with negative reinforcement, and the key role for Rac1 and p-cofilin in this process was shown. Is this also true for other forms of learning and memory, such as courtship memory? How does LTM forgetting occur in Drosophila?

At the same time, the picture of how higher nervous activity is based at the genetic and neuronal levels is now much clearer than it was a couple of decades ago. The rapid development of artificial intelligence may help us better understand how the brain works. At the same time, the modern ANN architectures take into account the basic principles of brain function. Productive cooperation between neurobiology and deep learning of ANNs will advance both disciplines.

## Figures and Tables

**Figure 1 ijms-26-11215-f001:**
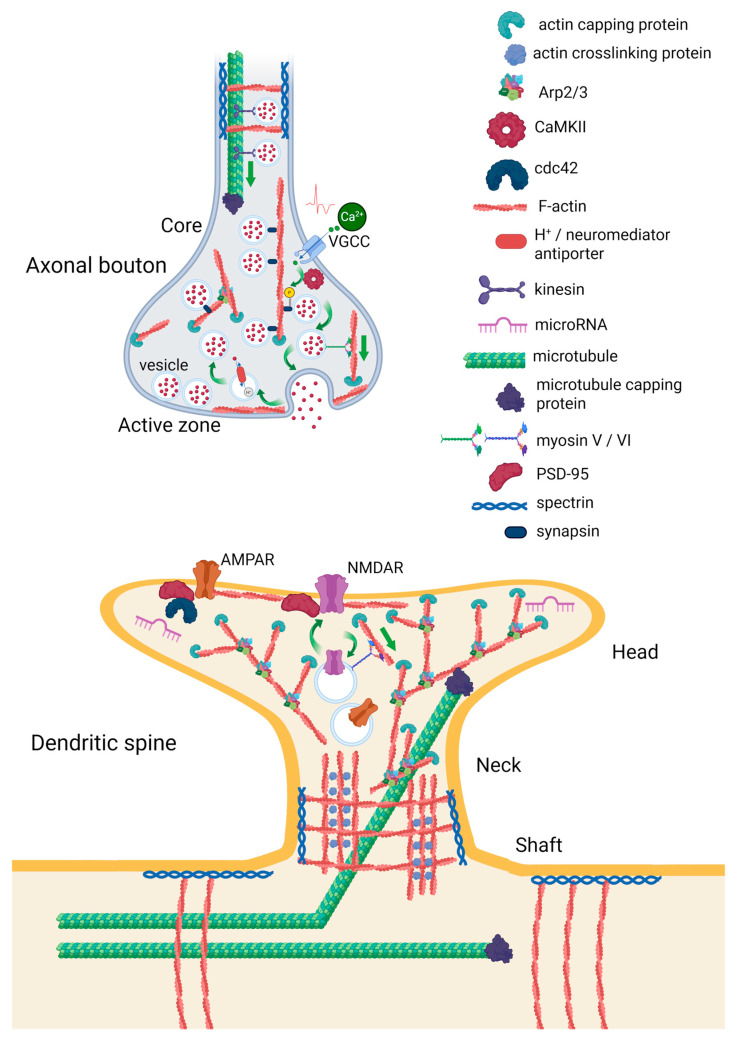
Organization of actin cytoskeleton in axonal boutons and dendritic spines. The illustration is prepared according to [25,26]. Created in BioRender. Zhuravlev, A. (2025) https://BioRender.com/rl8wxdw.

**Figure 2 ijms-26-11215-f002:**
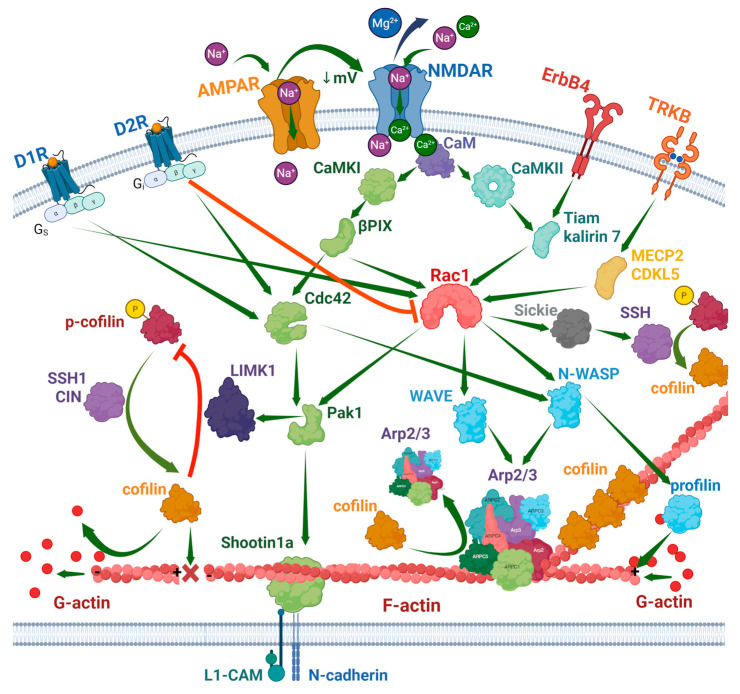
Signaling pathways of actin remodeling in neurons. Created in BioRender. Zhuravlev, A. (2025) https://BioRender.com/8m83p0s.

**Figure 3 ijms-26-11215-f003:**
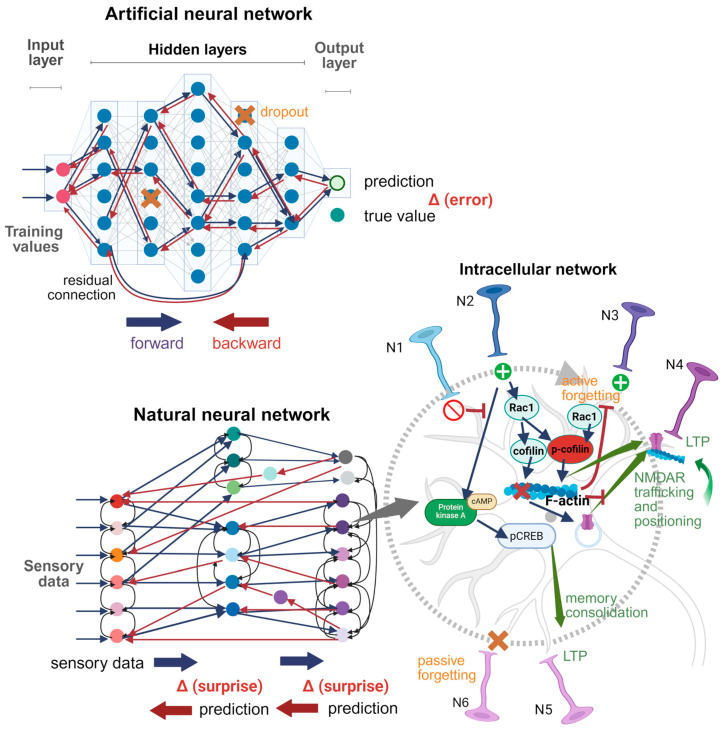
ANN and a natural neural network. Based on [195,196]. Created in BioRender. Zhuravlev, A. (2025) https://BioRender.com/gffauoe. Forward and backward connections are shown by blue and red, respectively. Collateral connections in the brain are shown by black arrows. Connections in ANN that are selectively strengthened after training are shown by a bold line. Intracellular network: N1–N6, input neurons. N1, inhibitory neuron; N2, excitatory neurons; N3, excitatory neuron, which output activity is blocked by active forgetting; N4, neuron, which synaptic efficacy is increased by protein synthesis-independent processes; N5, neuron, which synaptic efficacy is increased by protein synthesis-dependent processes; N6, neuron, which output activity is blocked by passive forgetting.

**Table 1 ijms-26-11215-t001:** Actin-remodeling factors and memory processes.

Protein Factor	Animal and Brain Area	Type of Learning/Memory	Effectors/Mechanism	Reference
Learning/Memory				
NMDAR	mouse hippocampus	spatial LTM	→ calcineurin → PIK3 → cofilin → translocation to spines (with the help of β-arrestin-2)	[121]
Rac1	mouse hippocampus	spatial learning, episodic-like memory	→ PAK, (→ LIMK -| cofilin ?), new spine stabilization	[87]
Rac1	rat BLA	auditory fear LTM reconsolidation	unknown	[120]
Rac1	mouse BLA	conditioned fear STM, LTM	unknown	[118]
Rac1	mouse hippocampus, presynaptic	spatial working memory	affects the distribution and morphology of synaptic vesicles	[119]
Rac1	mouse hippocampus, postsynaptic	contextual fear LTM	unknown	[119]
profilin	rat lateral amygdala	fear LTM	VASP, Arp2/3; stabilization of dendrite cytoskeleton	[125]
cAMP–PKA	mouse hippocampus	sleep-dependent object-place LTM	→ LIMK1 -| cofilin, spines stabilization	[117]
cofilin	mouse forebrain	spatial, aversive, and rewarded learning	→ AMPAR mobility	[100]
cofilin	mouse hippocampus	object-location STM	unknown	[122]
WRAP	mouse hippocampus	spatial LTM	affects spice density and synaptic plasticity	[123]
moesin	Drosophila MB γ neurons	courtship LTM	unknown	[135]
α-adducin	*C. elegans*, human	aversive olfactory STM and LTM (*C. elegans*), episodic memory (human)	actin capping	[126]
Forgetting				
Rac1	Drosophila MB	olfactory STM intrinsic forgetting	→ PAK → (LIMK1 -| ?) → cofilin	[140]
Rac1	mouse hippocampus	spatial memory forgetting	learning-evoked neurogenesis (?)	[160]
Rac1	mouse motor cortex	motor learning suppress	spine shrinkage	[158]
Rac1	mouse hippocampus	object recognition LTM forgetting	filopodia-like spines formation	[161]
Rac1	mouse lateral amygdala	auditory fear LTM suppress	→ PAK	[159]
Rac1	mouse BLA, astrocytes	fear LTM suppress	(LIMK1 -| ?)cofilin	[172]
Rac1	Drosophila γ MB neurons	olfactory ASM forgetting	→ SCAR/WAVE → Dia (→ linear actin?)	[142]
Cdc42	Drosophila MB	olfactory ARM forgetting	→ WASP → Arp2/3 (→ branched actin?)	[141,142]
MSI-1	*C. elegans* AVA interneuron	olfactory STM/LTM forgetting	-|Arp2/3, (reduced actin branching ?)	[146]
Dop1R2	Drosophila αα’ MB neurons	olfactory interference-based forgetting	→ Rac1 → PAK3 → (LIMK1 -| ?) cofilin	[152]
LIMK1-| cofilin	Drosophila MB	courtship STM forgetting	unknown	[156]
profilin	Drosophila (MB?)	immediate courtship memory impairment	unknown	[165,166]

Notes: →, activation; -|, inhibition; ?, supposed.

## Data Availability

No new data were created or analyzed in this study. Data sharing is not applicable to this article.

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
