# Peer review of "Neuronal Actin Remodeling and Its Role in Higher Nervous Activity"

_ijms, 2025, doi:10.3390/ijms262211215_

Round 1

Reviewer 1 Report

Comments and Suggestions for Authors

The manuscript by Aleksandr V. Zhuravlev, entitled “Neuronal Actin Remodeling and Its Role in Higher Nervous Activity,” describes the physiological role of actin filaments. The authors also discuss how these filaments contribute to memory retention and forgetting. The manuscript is well written, and incorporating the minor suggestions listed below will further enhance its clarity and overall quality.

  1. Line 221, “release from from ADP…..” In this line, the word “from” is repeated Please remove the extra “from”.
  2. I recommend including a paragraph on how actin remodeling occurs in Alzheimer’s disease in Section 7, “Actin Remodeling and Neuropathologies”. This would provide a more comprehensive view of the pathological relevance of actin dynamics in neurodegenerative disorders. Does it different from normal forgetting?
  3. The author may consider including a brief section on the potential effects of nootropic drugs on memory forgetting and actin cytoskeletal dynamics.
  4. I recommend incorporating the age factor when discussing actin dynamics and their impact on memory retention and forgetting, as aging is known to influence cytoskeletal plasticity and cognitive performance in the Section-5, “The role of actin cytoskeleton in learning and memory retention” and Section-6, “Actin cytoskeleton and memory forgetting”.

Author Response

Comments 1: Line 221, “release from from ADP…..” In this line, the word “from” is repeated Please remove the extra “from”.

Response 1: I have corrected the mistake.

Comments 2: I recommend including a paragraph on how actin remodeling occurs in Alzheimer’s disease in Section 7, “Actin Remodeling and Neuropathologies”. This would provide a more comprehensive view of the pathological relevance of actin dynamics in neurodegenerative disorders. Does it different from normal forgetting?

Response 2: The text describing actin remodeling processes in Alzheimer’s disease with the references has been added to the appropriate place: page 16 – 17, lines 666-686. The processes of spine loss in AD is reminiscent of what occurs in LIMK1-dependent forgetting.

Comments 3: The author may consider including a brief section on the potential effects of nootropic drugs on memory forgetting and actin cytoskeletal dynamics.

Response 3: The text with the references has been added to the appropriate place of Section 7: page 17, paragraph 3, lines 693-702.

Comments 4: I recommend incorporating the age factor when discussing actin dynamics and their impact on memory retention and forgetting, as aging is known to influence cytoskeletal plasticity and cognitive performance in the Section-5, “The role of actin cytoskeleton in learning and memory retention” and Section-6, “Actin cytoskeleton and memory forgetting”.

Response 4: The text with the references has been added to the appropriate place of Section 7: page 17, paragraph 4, lines 703-711; see also page 17, paragraph 5, lines 720-722.

Reviewer 2 Report

Comments and Suggestions for Authors

This is a fascinating review detailing the relationship between actin and memory. The text is extremely well written, although a bit wordy at some places, and written by someone with clear expertise in both neuroscience and actin dynamics. This review can be accepted when the author addresses the following:

  • The author paints a very detailed picture of the relationship between actin polymerization and dendritic spine formation. However, two things need to be addressed in this part:
    • The role of contractile proteins, specifically non-muscle myosin II, which is the main target of the RhoA/ROCK axis in dendritic spine formation. See for example the work from Alan Rick Horwitz’s group (doi:10.1371/journal.pone.0024149)
    • The difference between actin organization and dynamics in excitatory vs. inhibitory synapses (lines 164-166). The author mentions they are different, but this is an interesting enough topic to merit additional description.
  • A recent paper in JCB describes the role of non-muscle myosin II in presynaptic portals. Please discuss this.
  • While you are at it, please describe the adhesive clutch theory, which emanates from growth cone experiments, mainly by Paul Forscher. This pertains directly to actin dynamics in presynaptic portals.
  • Since the author refers to active forgetting, it feels like a missed opportunity to describe actin-related changes in diseases with a strong memory component, e.g. Alzheimer's disease. Please add a small section or paragraph on this.

Other than that, this is a sincerely excellent piece.

Author Response

Comments 1: The role of contractile proteins, specifically non-muscle myosin II, which is the main target of the RhoA/ROCK axis in dendritic spine formation. See for example the work from Alan Rick Horwitz’s group (doi:10.1371/journal.pone.0024149)

Response 1: The text with the references has been added to the appropriate place of Section 4: page 10, paragraph 2, lines 399-402.

Comments 2: The difference between actin organization and dynamics in excitatory vs. inhibitory synapses (lines 164-166). The author mentions they are different, but this is an interesting enough topic to merit additional description.

Response 2: The difference has been described in more detail: Section 2, page 5, paragraph 2, lines 175 – 179; see also page 5, paragraph 1, lines 155 – 161.

Comments 3: A recent paper in JCB describes the role of non-muscle myosin II in presynaptic portals. Please discuss this.

Response 3: The text with the references has been added to the appropriate place of Section 4: page 10, paragraph 2, lines 402-404.

Comments 4: While you are at it, please describe the adhesive clutch theory, which emanates from growth cone experiments, mainly by Paul Forscher. This pertains directly to actin dynamics in presynaptic portals.

Response 4: The text with the references describing processes of axon growth cone has been added to the appropriate place of Section 4: page 10, paragraph 1, lines 390-398.

Comments 5: Since the author refers to active forgetting, it feels like a missed opportunity to describe actin-related changes in diseases with a strong memory component, e.g. Alzheimer's disease. Please add a small section or paragraph on this.

Response 5: The text describing actin remodeling processes in Alzheimer’s disease with the references has been added to the appropriate place: page 16 – 17, lines 666-686.

Round 2

Reviewer 2 Report

Comments and Suggestions for Authors

No further comments